

# Identification and analysis of CYP450 genes from transcriptome of *Lonicera japonica* and expression analysis of chlorogenic acid biosynthesis related CYP450s

Xiwu Qi[1,2], Xu Yu[1,2], Daohua Xu[1], Hailing Fang[1,2], Ke Dong[3], Weilin Li[1,2] and Chengyuan Liang[1,2]

[1] Institute of Botany, Jiangsu Province and Chinese Academy of Sciences, Nanjing, China
[2] The Jiangsu Provincial Platform for Conservation and Utilization of Agricultural Germplasm, Nanjing, China
[3] Department of Biological Sciences, College of Natural Sciences, Seoul National University, Seoul, South Korea

Corresponding authors
Weilin Li, 1923872828@qq.com
Chengyuan Liang, liangcy618@aliyun.com

## ABSTRACT

**Background**. *Lonicera japonica* is an important medicinal plant that has been widely used in traditional Chinese medicine for thousands of years. The pharmacological activities of *L. japonica* are mainly due to its rich natural active ingredients, most of which are secondary metabolites. CYP450s are a large, complex, and widespread superfamily of proteins that participate in many endogenous and exogenous metabolic reactions, especially secondary metabolism. Here, we identified CYP450s in *L. japonica* transcriptome and analyzed CYP450s that may be involved in chlorogenic acid (CGA) biosynthesis.

**Methods**. The recent availability of *L. japonica* transcriptome provided opportunity to identify CYP450s in this herb. BLAST based method and HMM based method were used to identify CYP450s in *L. japonica* transcriptome. Then, phylogenetic analysis, conserved motifs analysis, GO annotation, and KEGG annotation analyses were conducted to characterize the identified CYP450s. qRT-PCR was used to explore expression patterns of five CGA biosynthesis related CYP450s.

**Results**. In this study, 151 putative CYP450s with complete cytochrome P450 domain, which belonged to 10 clans, 45 families and 76 subfamilies, were identified in *L. japonica* transcriptome. Phylogenetic analysis classified these CYP450s into two major branches, A-type (47%) and non-A type (53%). Both types of CYP450s had conserved motifs in *L. japonica*. The differences of typical motif sequences between A-type and non-A type CYP450s in *L. japonica* were similar with other plants. GO classification indicated that non-A type CYP450s participated in more molecular functions and biological processes than A-type. KEGG pathway annotation totally assigned 47 CYP450s to 25 KEGG pathways. From these data, we cloned two *LjC3Hs* (CYP98A subfamily) and three *LjC4Hs* (CYP73A subfamily) that may be involved in biosynthesis of CGA, the major ingredient for pharmacological activities of *L. japonica*. qRT-PCR results indicated that two *LjC3Hs* exhibited opposing expression patterns during the flower development and *LjC3H2* exhibited a similar expression pattern with CGA concentration measured by HPLC. The expression patterns of three *LjC4Hs* were quite different and the expression pattern of *LjC4H3* was quite similar with that of *LjC3H1*.

**Discussion**. Our results provide a comprehensive identification and characterization of CYP450s in *L. japonica*. Five CGA biosynthesis related *CYP450s* were cloned and their expression patterns were explored. The different expression patterns of two *LjC3Hs* and three *LjC4Hs* may be due to functional divergence of both substrate and catalytic specificity during plant evolution. The co-expression pattern of *LjC3H1* and *LjC4H3* strongly suggested that they were under coordinated regulation by the same transcription factors due to same *cis* elements in their promoters. In conclusion, this study provides insight into CYP450s and will effectively facilitate the research of biosynthesis of CGA in *L. japonica*.

# INTRODUCTION

Cytochrome P450 monooxygenases (CYP450s) are a large and complex superfamily which can be found in almost all living organisms (*Nelson, 1999*). Plant CYP450s are heme-containing enzymes that take part in a wide variety of reactions of both primary and secondary metabolism (*Kumar et al., 2014*), including the production of fatty acids, sterols, plant hormones, flavonoids, terpenoids, lignin, signaling molecules, and other biological molecules (*Schuler & Werck-Reichhart, 2003*).

*Lonicera japonica* Thunb. is a perennial evergreen vine belonging to the family Caprifoliaceae. *L. japonica* is a medicinal plant of great importance in traditional Chinese medicine that has been used for thousands of years (*Shang et al., 2011*). There are more than 500 traditional Chinese medicine prescriptions containing *L. japonica* (*Shang et al., 2011*). Modern pharmacological studies have indicated that the extracts of *L. japonica* possess many biological and pharmacological activities, such as anti-inflammatory, antiviral, antibacterial, antioxidant, hepato-protective, anti-tumor, and other activities (*Xiang et al., 2001*; *Yoo et al., 2008*).

The active compounds of *L. japonica* have been extensively studied. Essential oils (*Schlotzhauer, Pair & Horvat, 1996*), phenolic acids (*Lu, Jiang & Chen, 2004*), flavone (*Chen et al., 2005*), triterpenoid saponins (*Chai, Li & Li, 2005*), iridoilds and inorganic elements as the main compositions were isolated and identified in *L. japonica*. Among all these products, chlorogenic acid (CGA) is the major ingredient for pharmacological activities and its content is typically used as the main indicator of quality for evaluating *L. japonica* (*Chinese Pharmacopoeia Commission, 2010*).

As one of the most important secondary metabolites in plants, CGA is often used in medicines and foods for its high anti-oxidative activity (*Zucker & Levy, 1959*). The biosynthetic pathway of CGA has been investigated in many plants and is catalyzed by a series of enzymes (*Niggeweg, Michael & Martin, 2004*). Cinnamate 4-hydroxylase (C4H) and *p*-coumarate 3′-hydroxylase (C3H) are two CYP450s that participate in the two steps of hydroxylation in CGA biosynthetic pathway (*Gabriac et al., 1991*; *Schoch et al., 2001*). In *L. japonica*, a CYP98A subfamily gene encoding LjC3H was isolated and characterized.

By using heterologous expressed LjC3H *in vitro* assay, a recent study revealed that the recombinant protein was effective in converting *p*-coumaroylquinate to CGA (*Pu et al., 2013*). Two *C4Hs* belonging to the CYP73A subfamily were also cloned in *L. japonica*. Expression and activity analysis suggested that *LjC4H2* may be one of the critical genes that regulate CGA content in *L. japonica* (*Yuan et al., 2014*).

The studies of *L. japonica* have been focused on the identification of active compounds and pharmacological activity assays. In recent years, with the technological advancement in molecular biology, especially the development of next-generation sequencing technology, great progress has been made in the identification of active compounds involved in the biosynthesis processes in *L. japonica* (*Yuan et al., 2012*; *He et al., 2013*). In this study, bioinformatics tools were used to identify and analyze the *CYP450* genes based on transcriptome data of *L. japonica*. We identified two *LjC3Hs* and three *LjC4Hs* from the *CYP450* candidate genes, which including one previously reported *LjC3H* and two *LjC4Hs* genes. We further cloned the five *CYP450* genes and analyzed their transcriptional patterns in different developmental stages flowers. The results provided here will expand CYP450s information and could effectively facilitate CGA biosynthetic studies in *L. japonica*.

## MATERIALS AND METHODS

### Identification of CYP450 genes in *L. japonica*

The transcriptome data of *L. japonica* generated from different sequencing platforms including 454 GS-FLX, Illumina HiSeq2000, and Illumina GA II was downloaded from the NCBI SRA database with accession numbers SRR290309, SRR342027, SRR576924, SRR576925 and SRR766791. Four datasets were assembled and annotated. To identify putative *CYP450* genes, both Hidden Markov Model (HMM) method and BLAST method were used. For HMM method, P450.hmm file which represents the Hidden Markov Model of the cytochrome P450 family was initially downloaded from Pfam (http://pfam.xfam.org/), and then, HMMER3 software (*Eddy, 2011*) was used to search P450.hmm against *L. japonica* deduced amino acid database. For BLAST method, 19,047 full length plant CYP450 sequences were retrieved from UniProt (http://www.uniprot.org/). These sequences were used as queries to tblastn against *L. japonica* transcriptome assembly with an $E$-value cutoff of 1e−5. After filtering out the repeated results, the coding sequences of the resultant subjects were retrieved. Finally, results from the two methods were integrated and corrected manually. The identification methods were conducted for the four datasets of *L. japonica* and the results were also integrated and corrected. The corrected *L. japonica* CYP450s were further submitted to NCBI Conserved Domain Search (http://www.ncbi.nlm.nih.gov/Structure/cdd/wrpsb.cgi) to predict the conserved domain. Sequences with complete cytochrome P450 domain were selected for further analysis.

### Classification and characterization of *L. japonica* CYP450 genes

*L. japonica* CYP450s were classified into different families and subfamilies according to the sequence similarity using sequences from Cytochrome P450 Homepage as reference sequences. If the amino acid sequences of *L. japonica* CYP450s showed >40%, >55%, or

>95% sequence similarity with reference sequences, they were classified into the same family, subfamily, or allelic variant, respectively (*Nelson, 2009*).

The deduced amino acid sequences of *L. japonica* CYP450s were subjected to Multiple Expectation Maximization for Motif Elicitation (MEME, http://meme-suite.org/) (*Bailey et al., 2009*) analysis for identification of conserved motifs. Sequences of the four conserved CYP450 motifs including heme-binding region, PERF motif, K-helix region and I-helix region were extracted and then subjected to WEBLOGO (http://weblogo.berkeley.edu/) (*Crooks et al., 2004*) to create the sequence logos.

## Phylogenetic analysis of predicted CYP450 genes

A total of 63 representative sequences from plant CYP450 families were selected for phylogenetic analysis with 151 *L. japonica* CYP450 sequences. Specifically, CYP450 sequences whose functions had already been identified were preferentially selected. Multiple sequence alignment was performed using MUSCLE 3.6 software (*Edgar, 2004*). The result of alignment was imported to MEGA4 (*Tamura et al., 2007*) and phylogenetic analysis was performed. The phylogenetic tree was constructed using the Neighbor-Joining algorithm with the Poisson model and pairwise deletion. Bootstrap testing with 1,000 replications was used to test the phylogenetic tree. The Newick format file of bootstrap consensus tree was exported and then modified using EvolView (http://www.evolgenius.info/evolview/) (*Zhang et al., 2012*).

## Gene ontology and KEGG pathway analysis

Blast2GO (http://www.blast2go.com/) (*Conesa et al., 2005*) was used to perform Gene ontology (GO) annotation of *L. japonica* CYP450s. These predicted genes were functionally categorized according to three different criterions including cellular component, molecular function and biological process. The GO terms of all *L. japonica* CYP450s were extracted and subjected to Web Gene Ontology Annotation Plot (WEGO, http://wego.genomics.org.cn/cgi-bin/wego/index.pl) (*Ye et al., 2006*) to plot GO annotation results. KEGG annotation that maps the *L. japonica* CYP450s to possible KEGG pathway for biological interpretation of systemic functions was also conducted using Blast2GO.

## Extraction and quantification of CGA

The *L. japonica* used for this study was maintained at the Germplasm Nursery in Institute of Botany, Jiangsu Province and Chinese Academy of Sciences, Nanjing, Jiangsu Province. Flower buds and flowers samples for CGA and RNA extraction were collected at five stages: young alabastrum (YA, ≤1.5 cm), green alabastrum (GA, 2.0–3.0 cm), while alabastrum (WA, 3.2–4.4 cm), silvery flower (SF, about 5 cm), and golden flower (GF, about 5 cm). The extraction and quantification of CGA were conducted as described in Chinese Pharmacopoeia with minor modifications (*Chinese Pharmacopoeia Commission, 2010*). Briefly, dried buds or flowers were separately comminuted with a miler, and 0.2 g of each solid sample (40 mesh) was extracted with 25 mL of 50% aqueous methanol by ultrasonication (250 W, 35 kHz) for 30 min. After cooling to room temperature, the extracts were replenished to earlier weights with 50% aqueous methanol. Then, 5 mL of the extracts were diluted to 25 mL with 50% aqueous methanol and filtered with 0.45 μm

Millipore filter membranes. An Agilent 1200LC series HPLC system was used to analyze the CGA levels. Separations were performed on an Agilent TC-C18 reserved-phase column (5 $\mu$m, 250 mm × 4.6 mm) at 25 °C. The mobile phase was composed of acetonitrile-0.4% $H_3PO_4$ (13:87). The flow rate was 1 mL/min and fractions were monitored at 327 nm. Components were identified by comparison of the retention times of the eluting peaks to those of commercial standards under the same conditions.

### RNA extraction and qRT-PCR

Total RNA from five samples was extracted using RNAiso Plus (Takara, Tokyo, Otsu, Shiga, Japan) according to the manufacturer's instructions. RNA quality and concentration were measured using a ND-1000 UV spectrophotometer (Nanodrop Technologies, Wilmington, DE, USA). First-strand cDNA was synthesized using 3 $\mu$g of total RNA with M-MLV reverse transcriptase (Promega, USA) in a 25 $\mu$l reaction system. For quantitative real-time reverse transcriptional PCR (qRT-PCR), each reaction was prepared according to the manufacturer's instructions using SYBR® Premix Ex TaqTM II (Takara) and 2 $\mu$l of diluted cDNA as a template. The qRT-PCR reactions were conducted on the qTOWER2.2 Real Time PCR Systems (Analytik, Jena, Germany). The *L. japonica* actin gene was used as a control to normalize the relative expression levels of target genes. Gene-specific primers used for qRT-PCR were listed on Table S1. All results were representative of three independent experiments.

## RESULTS

### Identification and classification of CYP450 genes in *L. japonica*

Cytochrome P450 is one of the most massive gene superfamilies that is comprised of a number of families and subfamilies. In the present study, by integrating the results from different datasets and manual correction, we in total identified 151 putative CYP450s with complete cytochrome P450 domain in *L. japonica*. Among them, nine CYP450s had been previously reported and the other 142 CYP450s were identified here for the first time in *L. japonica*. Based on sequence similarity, we classified the 151 *CYP450* genes from *L. japonica* into 10 clans consisting of 45 families and 76 subfamilies (Table 1). Among them, the CYP71 clan, which represents the whole set of A-type *CYP450* genes, contains 71 genes belonging to 19 families (CYP71, CYP73, CYP75–CYP84, CYP89, CYP92, CYP93, CYP98, CYP701, CYP706, and CYP736). The non-A type *CYP450* genes of *L. japonica* contains the remaining 80 genes, which belongs to nine CYP clans (CYP51, 72, 74, 85, 86, 97, 710, 711, and 727) and 26 families (CYP51, CYP72, CYP714, CYP715, CYP721, CYP734, CYP749, CYP74, CYP85, CYP87, CYP88, CYP90, CYP707, CYP716, CYP722, CYP724, CYP728, CYP729, CYP86, CYP94, CYP96, CYP704, CYP97, CYP710, CYP711, and CYP727). The largest CYP family of *L. japonica* is CYP71 and CYP72, containing 17 and 18 members, respectively.

### Phylogenetic analysis of predicted CYP450s in *L. japonica*

Representative members of each plant CYP450 family were selected and used to conduct phylogenetic analysis with 151 CYP450s from *L. japonica*. The predicted CYP450s were

**Table 1  List of predicted CYP450s with complete cytochrome P450 domain from *L. japonica*.**

| Type | Clan | Family | Subfamily | Gene ID | Type | Clan | Family | Subfamily | Gene ID |
|------|------|--------|-----------|---------|------|------|--------|-----------|---------|
| non-A | 51 | CYP51 | CYP51G | m183961 | non-A | 72 | CYP72 | CYP72A | m61801 |
| non-A | 51 | CYP51 | CYP51G | m52657 | non-A | 72 | CYP72 | CYP72A | m25640 |
| A | 71 | CYP71 | CYP71B | m62714 | non-A | 72 | CYP72 | CYP72A | m206268 |
| A | 71 | CYP71 | CYP71D | m153867 | non-A | 72 | CYP72 | CYP72A | m132911 |
| A | 71 | CYP71 | CYP71D | m20042 | non-A | 72 | CYP72 | CYP72A | m20456 |
| A | 71 | CYP71 | CYP71D | m123612 | non-A | 72 | CYP72 | CYP72A | m161676 |
| A | 71 | CYP71 | CYP71D | m203354 | non-A | 72 | CYP72 | CYP72A | m16935 |
| A | 71 | CYP71 | CYP71D | m30084 | non-A | 72 | CYP72 | CYP72A | m216352 |
| A | 71 | CYP71 | CYP71D | m75810 | non-A | 72 | CYP72 | CYP72A | m196797 |
| A | 71 | CYP71 | CYP71D | m124427 | non-A | 72 | CYP72 | CYP72A | m21341 |
| A | 71 | CYP71 | CYP71D | m94101 | non-A | 72 | CYP72 | CYP72A | m194714 |
| A | 71 | CYP71 | CYP71D | m117052 | non-A | 72 | CYP72 | CYP72A | m76011 |
| A | 71 | CYP71 | CYP71AP | m198376 | non-A | 72 | CYP72 | CYP72A | m178417 |
| A | 71 | CYP71 | CYP71AU | m112981 | non-A | 72 | CYP72 | CYP72D | m62754 |
| A | 71 | CYP71 | CYP71AU | m12680 | non-A | 72 | CYP72 | CYP72D | m75640 |
| A | 71 | CYP71 | CYP71AU | m33704 | non-A | 72 | CYP714 | CYP714A | m189781 |
| A | 71 | CYP71 | CYP71BC | m197010 | non-A | 72 | CYP714 | CYP714E | m200247 |
| A | 71 | CYP71 | CYP71BE | m79469 | non-A | 72 | CYP714 | CYP714E | m125702 |
| A | 71 | CYP71 | CYP71BG | m82900 | non-A | 72 | CYP714 | CYP714E | m205273 |
| A | 71 | CYP73 | CYP73A | m177245 | non-A | 72 | CYP714 | CYP714E | m19972 |
| A | 71 | CYP73 | CYP73A | m13469 | non-A | 72 | CYP714 | CYP714E | m17561 |
| A | 71 | CYP73 | CYP73A | m8810 | non-A | 72 | CYP715 | CYP715A | m34769 |
| A | 71 | CYP75 | CYP75B | m13120 | non-A | 72 | CYP721 | CYP721A | m85505 |
| A | 71 | CYP76 | CYP76A | m204536 | non-A | 72 | CYP734 | CYP734A | m842 |
| A | 71 | CYP76 | CYP76A | m184619 | non-A | 72 | CYP749 | CYP749A | m139970 |
| A | 71 | CYP76 | CYP76A | m18954 | non-A | 74 | CYP74 | CYP74A | m4277 |
| A | 71 | CYP76 | CYP76A | m155830 | non-A | 74 | CYP74 | CYP74A | m50137 |
| A | 71 | CYP76 | CYP76A | m197465 | non-A | 74 | CYP74 | CYP74A | m67219 |
| A | 71 | CYP76 | CYP76B | m31162 | non-A | 74 | CYP74 | CYP74B | m52145 |
| A | 71 | CYP76 | CYP76B | m1881 | non-A | 74 | CYP74 | CYP74B | m19408 |
| A | 71 | CYP76 | CYP76B | m156655 | non-A | 85 | CYP85 | CYP85A | m206529 |
| A | 71 | CYP76 | CYP76B | m122126 | non-A | 85 | CYP85 | CYP85A | m42514 |
| A | 71 | CYP76 | CYP76Y | m139737 | non-A | 85 | CYP87 | CYP87D | m191193 |
| A | 71 | CYP76 | CYP76Y | m19896 | non-A | 85 | CYP88 | CYP88A | m84626 |
| A | 71 | CYP77 | CYP77A | m194721 | non-A | 85 | CYP90 | CYP90A | m21741 |
| A | 71 | CYP77 | CYP77B | m148606 | non-A | 85 | CYP90 | CYP90B | m8267 |
| A | 71 | CYP78 | CYP78A | m187124 | non-A | 85 | CYP90 | CYP90C | m119587 |
| A | 71 | CYP78 | CYP78A | m152788 | non-A | 85 | CYP90 | CYP90D | m88205 |
| A | 71 | CYP79 | CYP79D | m32635 | non-A | 85 | CYP707 | CYP707A | m212742 |
| A | 71 | CYP79 | CYP79D | m230122 | non-A | 85 | CYP707 | CYP707A | m213600 |
| A | 71 | CYP80 | CYP80C | m37356 | non-A | 85 | CYP707 | CYP707A | m47109 |

**Table 1** (*continued*)

| Type | Clan | Family | Subfamily | Gene ID | Type | Clan | Family | Subfamily | Gene ID |
|------|------|--------|-----------|---------|------|------|--------|-----------|---------|
| A | 71 | CYP81 | CYP81B | m211982 | non-A | 85 | CYP707 | CYP707A | m35702 |
| A | 71 | CYP81 | CYP81C | m12729 | non-A | 85 | CYP707 | CYP707A | m17557 |
| A | 71 | CYP81 | CYP81E | m131282 | non-A | 85 | CYP716 | CYP716A | m191349 |
| A | 71 | CYP81 | CYP81E | m61839 | non-A | 85 | CYP716 | CYP716A | m57776 |
| A | 71 | CYP81 | CYP81E | m61297 | non-A | 85 | CYP716 | CYP716A | m12551 |
| A | 71 | CYP82 | CYP82C | m99205 | non-A | 85 | CYP716 | CYP716C | m77065 |
| A | 71 | CYP82 | CYP82C | m76311 | non-A | 85 | CYP716 | CYP716C | m23342 |
| A | 71 | CYP82 | CYP82D | m169021 | non-A | 85 | CYP716 | CYP716D | m141170 |
| A | 71 | CYP82 | CYP82D | m39884 | non-A | 85 | CYP716 | CYP716E | m153199 |
| A | 71 | CYP82 | CYP82D | m211151 | non-A | 85 | CYP716 | CYP716E | m200248 |
| A | 71 | CYP82 | CYP82U | m215270 | non-A | 85 | CYP722 | CYP722A | m120593 |
| A | 71 | CYP82 | CYP82U | m56636 | non-A | 85 | CYP722 | CYP722C | m202676 |
| A | 71 | CYP82 | CYP82U | m82069 | non-A | 85 | CYP724 | CYP724A | m206239 |
| A | 71 | CYP82 | CYP82U | m61602 | non-A | 85 | CYP728 | CYP728B | m166264 |
| A | 71 | CYP83 | CYP83F | m86843 | non-A | 85 | CYP729 | CYP729A | m77833 |
| A | 71 | CYP83 | CYP83F | m64245 | non-A | 86 | CYP86 | CYP86A | m71018 |
| A | 71 | CYP83 | CYP83F | m109275 | non-A | 86 | CYP86 | CYP86A | m6298 |
| A | 71 | CYP83 | CYP83F | m91426 | non-A | 86 | CYP86 | CYP86C | m120347 |
| A | 71 | CYP84 | CYP84A | m176881 | non-A | 86 | CYP94 | CYP94A | m145786 |
| A | 71 | CYP84 | CYP84A | m124488 | non-A | 86 | CYP94 | CYP94B | m38093 |
| A | 71 | CYP89 | CYP89A | m131845 | non-A | 86 | CYP94 | CYP94C | m59371 |
| A | 71 | CYP92 | CYP92A | m14848 | non-A | 86 | CYP94 | CYP94C | m6650 |
| A | 71 | CYP92 | CYP92B | m61326 | non-A | 86 | CYP94 | CYP94D | m102827 |
| A | 71 | CYP93 | CYP93B | m79556 | non-A | 86 | CYP94 | CYP94D | m100765 |
| A | 71 | CYP98 | CYP98A | m184946 | non-A | 86 | CYP94 | CYP94F | m175121 |
| A | 71 | CYP98 | CYP98A | m43608 | non-A | 86 | CYP96 | CYP96A | m794 |
| A | 71 | CYP701 | CYP701A | m27329 | non-A | 86 | CYP96 | CYP96A | m21366 |
| A | 71 | CYP701 | CYP701A | m150262 | non-A | 86 | CYP704 | CYP704A | m94230 |
| A | 71 | CYP706 | CYP706C | m115920 | non-A | 97 | CYP97 | CYP97A | m56546 |
| A | 71 | CYP736 | CYP736A | m18282 | non-A | 97 | CYP97 | CYP97B | m17072 |
| A | 71 | CYP736 | CYP736A | m135731 | non-A | 97 | CYP97 | CYP97C | m3461 |
| A | 71 | CYP736 | CYP736A | m182725 | non-A | 710 | CYP710 | CYP710A | m92981 |
| non-A | 72 | CYP72 | CYP72A | m51504 | non-A | 711 | CYP711 | CYP711A | m201472 |
| non-A | 72 | CYP72 | CYP72A | m55535 | non-A | 727 | CYP727 | CYP727B | m144680 |
| non-A | 72 | CYP72 | CYP72A | m11850 | | | | | |

classified into two major branches, A-type (47%) and non-A type (53%) (Fig. 1). There were 10 clans in plants CYP450s. Four clans contained multiple families and were designated by their lowest-numbered family members, CYP71, CYP72, CYP85 and CYP86. The other six clans were designated by their only family, CYP51, CYP74, CYP97, CYP710, CYP711 and CYP727. In *L. japonica*, all 10 clans were identified. Genes belonging to same clan clustered as one clade. For example, the 72 clan, which comprised 28 CYP450s belonged to six families, were clustered as one clade with the eight representative CYP450s. The CYP71
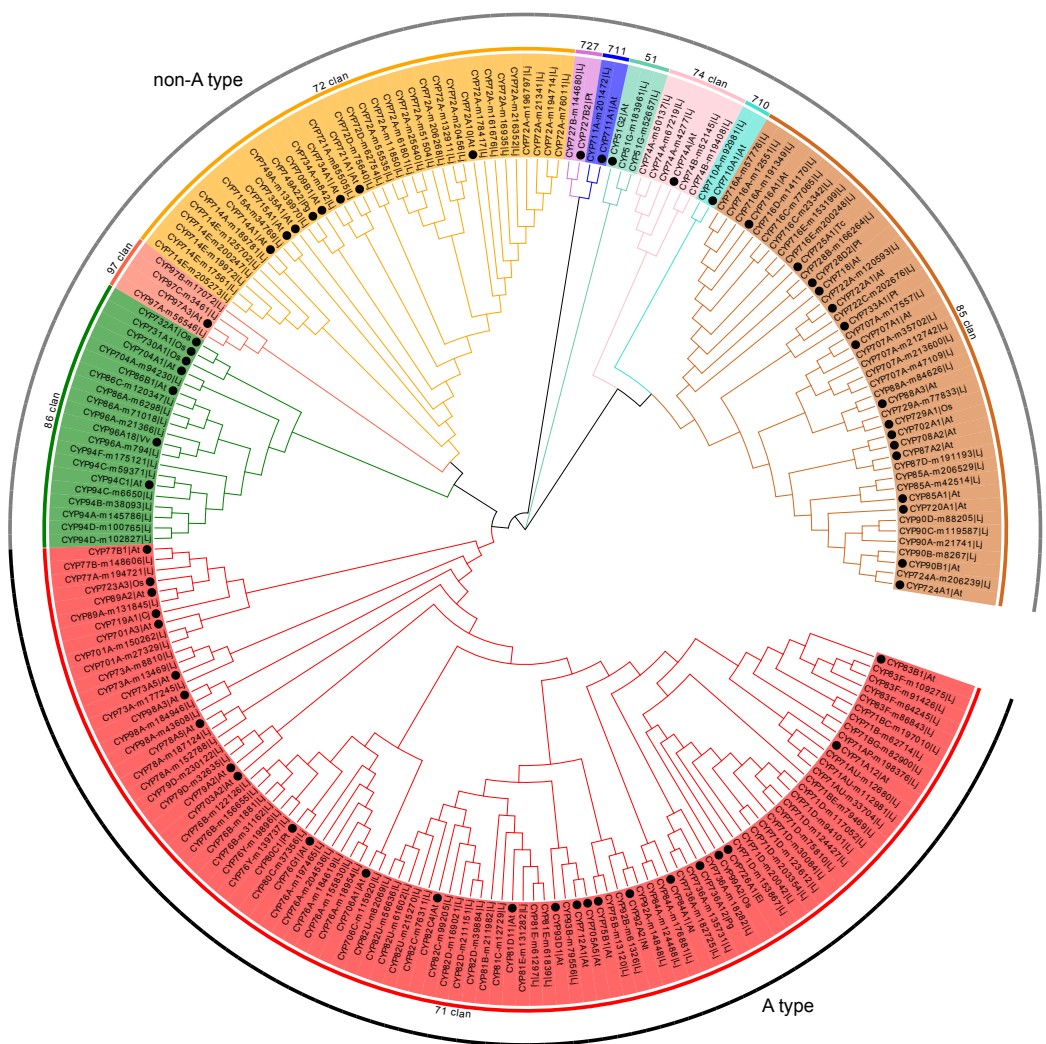

**Figure 1** Phylogenetic analysis of predicted CYP450s in *L. japonica* and the representative members of CYP450 families.

clan that comprised 71 members belonging to 19 families was the largest clan. Three clans, CYP710, CYP711 and CYP727, had only one member identified for each clan.

## Conserved motifs analysis of *L. japonica* CYP450s

Plant CYP450s shared some typical conserved motifs including heme-binding region, PERF motif, K-helix region and I-helix region, which were important for catalytic activities (*Paquette, Jensen & Bak, 2009*). The *L. japonica* CYP450s were divided into A-type and non-A type according to phylogenetic analysis. The deduced amino acid sequences were subjected to MEME to analyze the conserved motifs. The consensus sequences of the heme-binding region, also known as "P450 signature", were "PFGXGRRXCPG" and "XFXXGXRXCXG" for A-type and non-A type CYP450s, respectively (Fig. 2). The cysteine residues in this motif of two types of CYP450s were universally conserved, which links the heme iron to the apoprotein. The consensus sequences of the PERF motif were
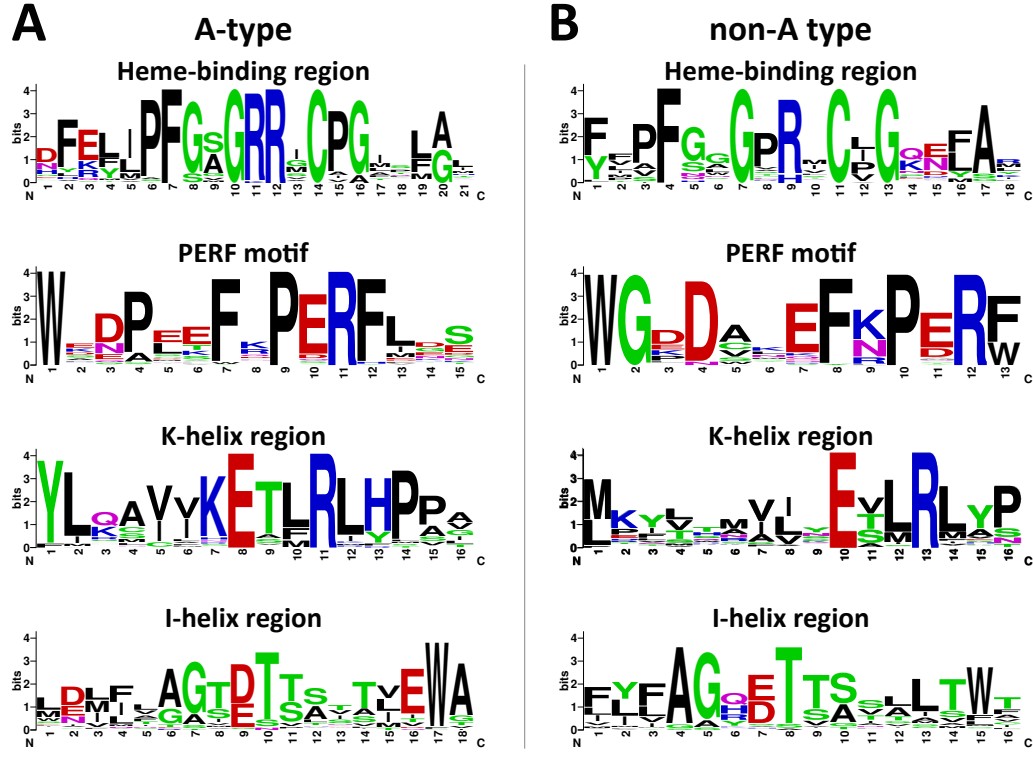

**Figure 2** Weblogos of conserved motifs identified in A-type (A) and non-A type (B) CYP450s from *L. japonica.*

also different for two types of CYP450s in *L. japonica*, which are "PERF" for A-type and "PXRX" for non-A type. The R residues in the PERF motif and E and R residues in the K-helix region were universally conserved, which form a salt bridge that has been proposed to be involved in locking the Cys-pocket in position and assuring the stable association of heme with the protein. The threonine residues in the I-helix region which is involved in oxygen activation was highly conserved in both A-type and non-A type CYP450s. In general, sequences of the typical motifs were conserved in *L. japonica* CYP450s, and the differences between A-type and non-A type CYP450s in *L. japonica* were similar with other plants (*Chen et al., 2014*).

## Gene ontology classification of *L. japonica* CYP450s

Gene ontology (GO) is a classification system for standardized gene functions which classifies genes into three main independent GO categories: cellular component, molecular function and biological process. In this study, GO assignments were conducted to classify the functions of CYP450s from *L. japonica* using Blast2GO. Results indicated that all 151 CYP450s were mapped to one or more GO terms, of which 145 were assigned to the "cellular component", 151 to the "molecular function", and 151 to the "biological process" (Fig. 3). Of these categories, cell, binding, catalytic, and metabolic process were the largest subcategories. Comparison of the GO classification between the A-type and non-A type CYP450s, we found that non-A type CYP450s participated in more molecular

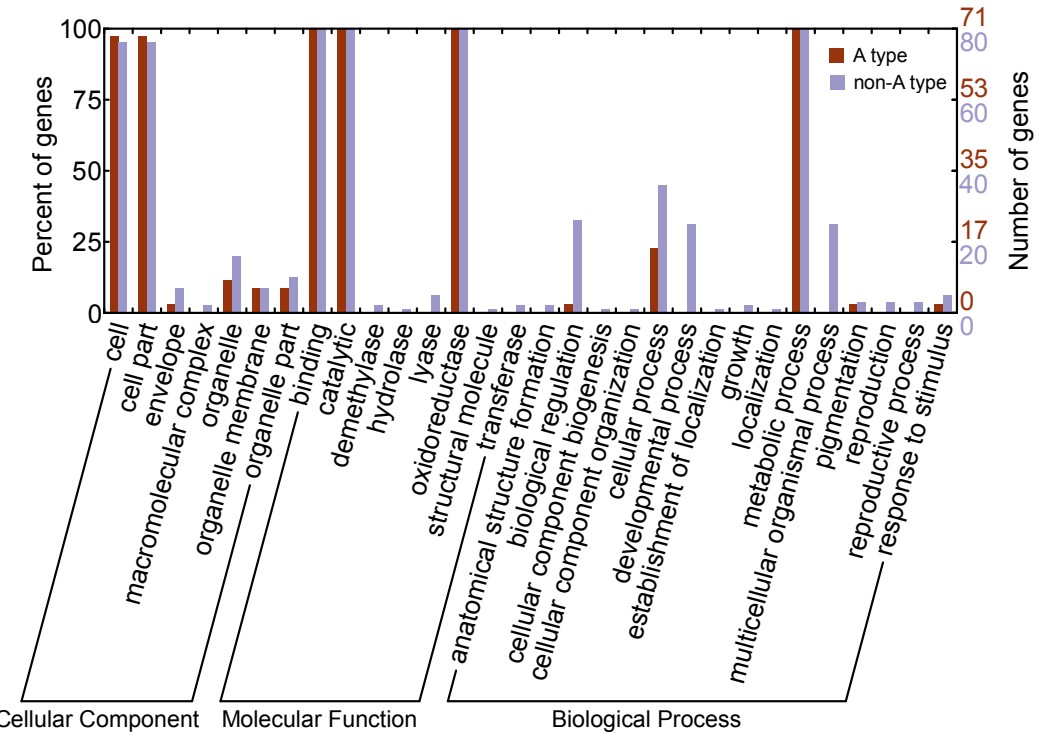

**Figure 3  Gene ontology annotation of A-type and non-A type CYP450s in *L. japonica*.**

functions and biological processes than A-type. For example, GO terms of non-A type CYP450s in molecular function category included demethylase, hydrolase, lyase, and transferase; however, no A-type CYP450s was assigned to these subcategories. In biological process category, non-A type CYP450s participated in more biological processes than A-type, including anatomical structure formation, cellular component organization, developmental process, establishment of localization, growth, localization, multicellular organismal process, and reproduction. The GO annotation provided a valuable clue to investigate the functions of CYP450s in *L. japonica*.

### KEGG pathway analysis of *L. japonica* CYP450s

In order to further understand the biological functions of CYP450s in *L. japonica*, pathway-based analysis was performed. Given that a CYP450 could be assigned to one or more KEGG pathways as well as GO terms, 47 (31.1%) CYP450s were totally assigned to 25 KEGG pathways (Fig. 4). The 25 pathways could be mainly grouped into six classes, including lipid metabolism, amino acid metabolism, metabolism of cofactors and vitamins, metabolism of terpenoids and polyketides, biosynthesis of other secondary metabolites, and xenobiotics biodegradation and metabolism. In the class of 'biosynthesis of other secondary metabolites', after removing duplicate hits, ten CYP450s (CYP73A-m13469, CYP73A-m177245, CYP73A-m8810, CYP75B-m13120, CYP76A-m155830, CYP78A-m152788, CYP93B-m79556, CYP98A-m184946, CYP98A-m43608 and CYP736A-m18282) were found to be involved in the biosynthesis of phenolic compounds

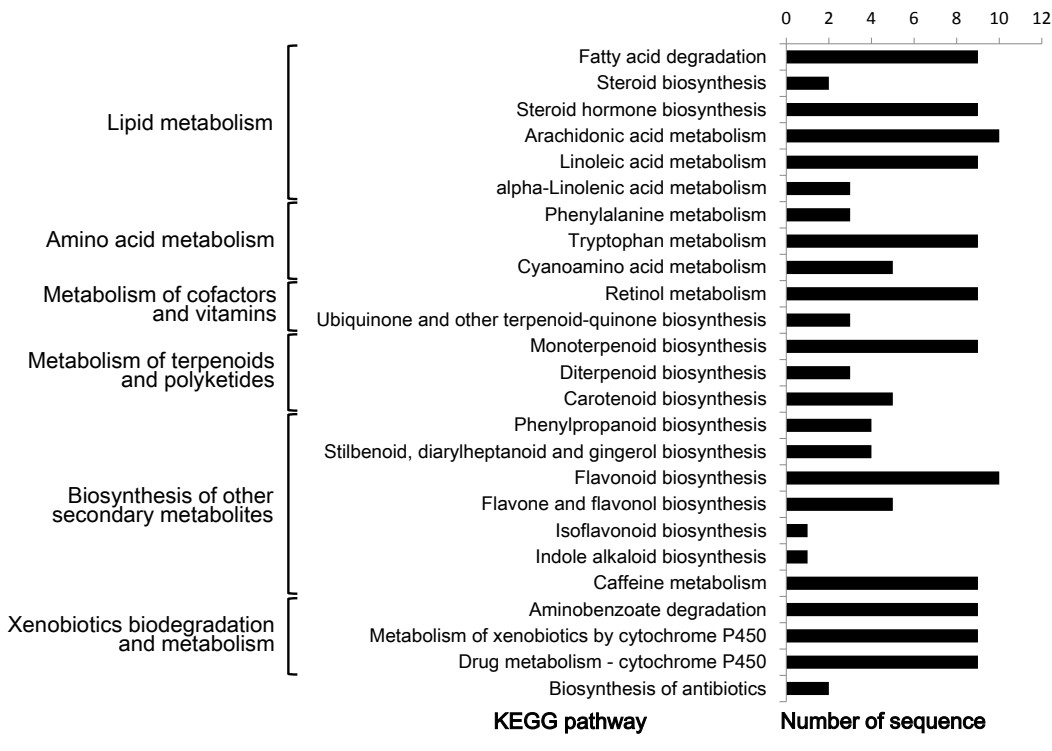

**Figure 4  KEGG pathway analysis of predicted CYP450s in *L. japonica*.**

including phenylpropanoid, stilbenoid, diarylheptanoid and gingerol, flavonoid, flavone and flavonol, and isoflavonoid. All ten CYP450s belonged to CYP71 clan. In the class of 'metabolism of terpenoids and polyketides', nine CYP450s (CYP72A-m132911, CYP72A-m20456, CYP72A-m206268, CYP72D-m62754, CYP72D-m75640, CYP714A-m189781, CYP714E-m17561, CYP714E-m205273 and CYP734A-m842) were found to be involved in 'monoterpenoid biosynthesis', all of which belonged to CYP72 clan. Three CYP450s (CYP701A-m150262, CYP701A-m27329 and CYP728B-m166264) were found to be involved in 'diterpenoid biosynthesis', among them, two belonged to CYP71 clan and one belonged to CYP85 clan. Five CYP450s (CYP707A-m213600, CYP707A-m35702, CYP707A-m47109, CYP707A-m212742 and CYP728B-m166264) were found to be involved in 'carotenoid biosynthesis', all of which belonged to CYP85 clan.

## CYP450s involved in CGA biosynthesis

CGA is the most major active ingredient in *L. japonica* and the biosynthetic pathway of CGA has been investigated in many plants. In CGA biosynthetic pathway, C4H and C3H are the two CYP450-encoded enzymes that participate in the two steps of hydroxylation. In the present study, three *C4H* and two *C3H* genes were identified and cloned from *L. japonica*. Among them, two *LjC4Hs* and one *LjC3H* have been previously reported. The newly identified *C4H* and *C3H* were designated as '*LjC4H3*' (GenBank accession number: KX845341) and '*LjC3H2*' (GenBank accession number: KX845342), respectively. The *C4Hs* belonged to CYP73A subfamily and *C3Hs* belonged to CYP98A subfamily. Phylogenetic

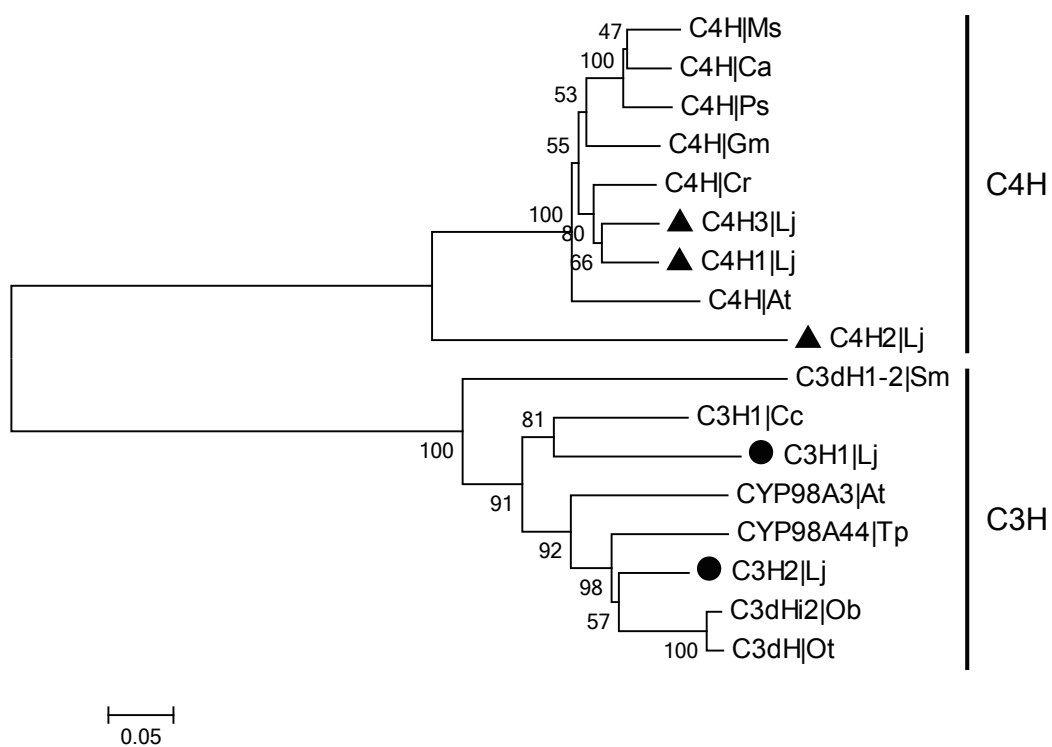

**Figure 5 Phylogenetic analysis of C3Hs and C4Hs from *L. japonica* and other plants.** LjC3Hs were labeled by black dots and LjC4Hs were labeled by black triangles. Protein sequences were downloaded from UniProt with accession numbers as follows: C3H1|Cc (A4ZKM5), CYP98A3|At (O22203), C3dH1-2|Sm (D8SCG3), C3dHi2|Ob (Q8L5H7), CYP98A44|Tp (C9EGT6), C3dH|Ot (T1NXG3), C4H|At (P92994), C4H|Cr (P48522), C4H|Ps (Q43067), C4H|Ms (P37114), C4H|Ca (O81928), C4H|Gm (Q42797).

analysis indicated that two clades were clustered for C4Hs and C3Hs from *L. japonica* and other plants (Fig. 5).

Because CGA was mainly accumulated in flower bud of *L. japonica*, buds and flowers in different developmental stages were selected to explore the relationship of *C4H* and *C3H* expressions and CGA contents. HPLC analysis was used to measure CGA concentrations in different developmental stages of buds and flowers. As shown in Fig. 6, the percentage of CGA contents decreased during the flower development. Nevertheless, with the increase of bud or flower weights, the total CGA contents increased from young alabastrum (YA) to while alabastrum (WA) stage and reached peak at the WA stage. After flowering, the total CGA contents decreased quickly during flower development. Furthermore, qRT-PCR was conducted to analyze the transcriptional levels of CGA biosynthetic genes in the different developmental stages of buds and flowers, including the five CYP450s identified in this study. The two *LjC3Hs* exhibited opposing expression patterns, the transcriptional levels of *LjC3H1* increased but that of *LjC3H2* decreased during the flower development (Fig. 7). The expression patterns of three *LjC4Hs* were quite different and the relative expression levels of *LjC4H3* was obviously higher than those of the other two (Fig. 7). Interestingly, the expression patterns of *LjPAL1*, *LjC4H3*, *LjC3H1* and *LjHQT* were quite similar, which

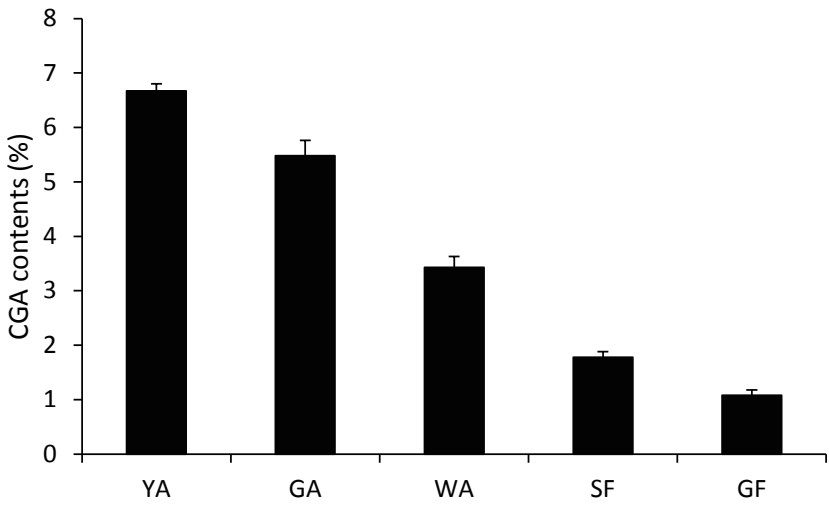

**Figure 6** **CGA contents of buds and flowers in different developmental stages *L. japonica.*** YA-young alabastrum, GA-green alabastrum, WA-white alabastrum, SF-silvery flower, and GF-golden flower.

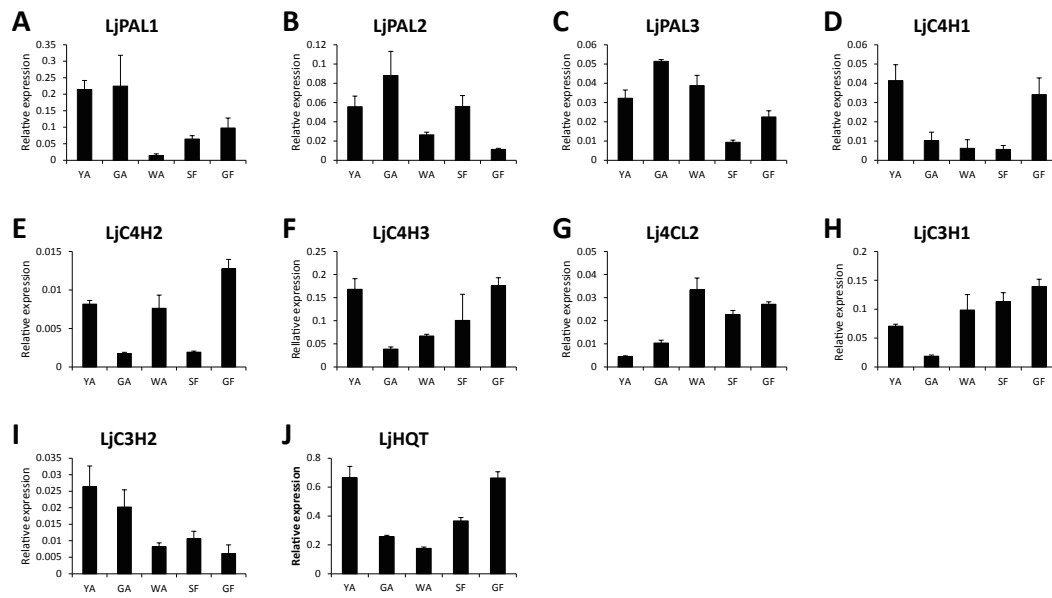

**Figure 7** **Transcriptional analyses of CGA biosynthetic pathway genes in buds and flowers of *L. japonica* at different developmental stages.**

exhibited a trend of decreasing first and then increasing. Considering the gene expressions with CGA contents, only *LjC3H2* exhibited a similar pattern with CGA concentrations.

## DISCUSSION

*L. japonica* is an important medicinal plant that has been widely used in traditional Chinese medicine for thousands of years. The pharmacological activities of this medicinal plant are mainly due to its rich natural active ingredients, most of which are secondary

metabolites. CYP450s are a large, complex, and widespread superfamily that participate in many metabolic reactions, especially secondary metabolism. The identification and characterization of *CYP450s* in *L. japonica* will effectively facilitate the study of natural active compounds biosynthesis. In this study, we identified 151 putative CYP450s with complete cytochrome P450 domain from transcriptome data of *L. japonica*. According to the classification criteria, the 151 CYP450s were classified into 10 clans consisting of 45 families and 76 subfamilies. Next, we conducted phylogenetic analysis, conserved motifs analysis, GO annotation, and KEGG annotation to characterize the identified CYP450s. As mentioned above, nine CYP450s have been previously reported in *L. japonica*, which were also identified among the 151 CYP450s of this study. These results indicated that the identified CYP450s from the *L. japonica* transcriptome data in this study were quite comprehensive.

The evolution of plant CYP450s can be divided into three major groups: CYP450s involved in sterol and carotenoid biosynthesis were the most ancient, CYP450s involved in adaptation to land environment were the next oldest, and CYP450s involved in biosynthesis of plant secondary metabolites were the most recent to evolve (*Morant et al., 2007*; *Nelson et al., 2008*). In this study, ten CYP450s (CYP73A-m13469, CYP73A-m177245, CYP73A-m8810, CYP75B-m13120, CYP76A-m155830, CYP78A-m152788, CYP93B-m79556, CYP98A-m184946, CYP98A-m43608, and CYP736A-m18282) were annotated to participate in the biosynthesis of phenolic compounds, a most common type of secondary metabolite in plants, including phenylpropanoid, stilbenoid, flavonoid, and isoflavonoid. All ten CYP450s belonged to CYP71 clan. As earlier reported, the most recently evolved CYP450 group comprises the highly proliferated clan 71. This clan includes CYP450s involved in the biosynthesis of the majority of plant secondary metabolites involved in adaptation to abiotic and biotic stress (*Morant et al., 2007*), with which our present findings are in agreement. Five CYP450s (CYP707A-m213600, CYP707A-m35702, CYP707A-m47109, CYP707A-m212742, and CYP728B-m166264) were found to be involved in carotenoid biosynthesis, all of which belonged to the CYP85 clan. These CYP450s belonged to the oldest group with a function that preceded the colonization of land by plants (*Morant et al., 2007*).

CGA is the major active ingredient in *L. japonica*, and the biosynthetic pathway of CGA has been investigated in many plants. In CGA biosynthetic pathway, C4H and C3H are two CYP450 encoded enzymes that participate in the two steps of hydroxylation (*Gabriac et al., 1991*; *Schoch et al., 2001*). In *L. japonica*, a gene encoding LjC3H has been isolated and characterized by *Pu et al. (2013)*, and was identified as CYP98A subfamily member. *In vitro* assay using heterologous expressed LjC3H revealed that the recombinant protein was effective in converting *p*-coumaroylquinate to CGA. Southern blotting suggested that the gene was present in the genome in two copies, but unfortunately, only one copy of *LjC3H* was obtained. In this study, two *LjC3Hs* were identified and cloned from *L. japonica*, both of which belonged to the CYP98A subfamily. Among the two *LjC3Hs*, one was same as the *LjC3H* reported by *Pu et al. (2013)*, the other is a newly identified gene and is hereby designated *LjC3H2*. These results suggested that the newly identified *LjC3H2* was the other copy of *LjC3H* in the genome of *L. japonica*. Two *C4Hs* were also cloned in *L. japonica*

by *Yuan et al. (2014)*, which belonged to the CYP73A subfamily. Expression and activity analysis suggested that *LjC4H2* may be one of the critical genes that regulate CGA content in *L. japonica*. In our study, three *C4Hs* were identified and cloned from *L. japonica*, including the previously reported two genes. The newly identified *LjC4H* was designated as *LjC4H3*, which showed high degree of sequence homology with *LjC4H1*. Phylogenetic analysis showed that LjC4H1 and LjC4H3 clustered to one clade. This result suggested that these two genes may be generated by recent gene duplication.

In the present study, the expression patterns of two *LjC3Hs* and three *LjC4Hs* were quite different during the flower development. This phenomenon that different members of the same family exhibit different expression patterns during development was also observed in other plants (*Bi et al., 2011*; *Qi et al., 2014*), which might be caused by functional divergence of both substrate and catalytic specificity during plant evolution (*Helariutta et al., 1996*; *Xu et al., 2009*). Considering the gene expressions with CGA contents, only *LjC3H2* exhibited a similar pattern with CGA concentrations in our study. This result was similar with that of coffee (*Lepelley et al., 2007*). In coffee, transcriptional levels of CGA biosynthetic genes and CGA contents were measured during grain development and *C3H1* showed a similar expression pattern with CGA concentrations. Both the CGA concentrations and *C3H* expression pattern were similar with those of *L. japonica*, respectively. However, in this study, the expression patterns of *LjC3H1* and three *LjC4Hs* were inconsistent with CGA contents during flower development. The reason for this phenomenon could be that C3H and C4H not only participated in CGA biosynthesis, but were also involved in other metabolites. The product catalyzed by C4H was a common precursor in phenylpropanoid metabolism, including flavonoids, anthocyanins, condensed tannins, and isoflavonoids (*Winkel-Shirley, 2001*). C3H was also a key enzyme in lignin biosynthesis (*Boerjan, Ralph & Baucher, 2003*). It is likely that the complexity of the metabolic pathways led to the inconsistency between gene expressions and product contents.

In this study, the expression patterns of *LjPAL1*, *LjC4H3*, *LjC3H1* and *LjHQT* were quite similar during flower development. The co-expression patterns of these four genes strongly suggested that they were under coordinated regulation by the same transcription factors due to similar *cis* elements in their promoters (*Bi et al., 2011*). In apple, anthocyanin biosynthetic genes including *CHS*, *CHI*, *F3H*, *DFR*, *LDOX* and *UFGT* showed similar expression patterns during fruit development, which were coordinately regulated by a MYB transcription factor, *MdMYB10* (*Espley et al., 2007*). Fruit-specific ectopic expression of *AtMYB12* in tomato led to upregulation of all biosynthetic genes required for the production of flavonols and their derivatives, including *PAL*, *C4H*, *4CL*, *CHS*, *CHI*, *F3H*, *F3'H*, *FLS*, *ANS*, *C3H*, *HCT*, *HQT*, *GT*, and *RT*; and, in addition, led to the increase of flavonols and their derivatives (*Luo et al., 2008*). In pine and eucalyptus, xylem-associated MYB transcription factors could bind to the AC elements and activate the transcription of the lignin biosynthetic genes (*Patzlaff et al., 2003*; *Goicoechea et al., 2005*). Moreover, the rice genome sequence analysis revealed that ACII motif existed in the promoters of many lignin biosynthetic genes, including *PAL*, *4CL*, *C4H*, *C3H*, *CCoAOMT*, *CCR*, and *CAD*, suggesting that they were under coordinated regulation by the same transcription factors (*Bi et al., 2011*).

## CONCLUSIONS

In this study, we identified 151 putative CYP450s with complete cytochrome P450 domain in *L. japonica* transcriptome, 142 of which were identified here for the first time. According to the classification criteria, the 151 CYP450s were classified into 10 clans consisting of 45 families and 76 subfamilies. Next, we conducted phylogenetic analysis, conserved motifs analysis, GO annotation, and KEGG annotation to characterize the identified CYP450s. From these data, we cloned two *LjC3Hs* (CYP98A subfamily) and three *LjC4Hs* (CYP73A subfamily) genes that may be involved in biosynthesis of CGA, including the newly identified *LjC3H2* and *LjC4H3*. Furthermore, qRT-PCR and HPLC results indicated that only *LjC3H2* exhibited a similar expression pattern with CGA concentration. Different members of the same family exhibited different expression patterns during development that may be due to functional divergence of both substrate and catalytic specificity during plant evolution. The co-expression pattern of *LjPAL1*, *LjC4H3*, *LjC3H1* and *LjHQT* strongly suggested that they were under coordinated regulation by the same transcription factors due to same *cis* elements in their promoters. In conclusion, this study provides insight into CYP450s and will effectively facilitate the research of biosynthesis of CGA in *L. japonica*.

### Funding

The work was supported by the National Natural Science Foundation of China (31500249), Six Talent Peaks Project in Jiangsu Province (2015-NY-032), the Natural Science Foundation of Jiangsu Province (BK20161381), and the Foundation of Jiangsu Key Laboratory for the Research and Uti1ization of Plant Resources (SQ201401). The funders had no role in study design, data collection and analysis, decision to publish, or preparation of the manuscript.

### Grant Disclosures

The following grant information was disclosed by the authors:
National Natural Science Foundation of China: 31500249.
Six Talent Peaks Project in Jiangsu Province: 2015-NY-032.
Natural Science Foundation of Jiangsu Province: BK20161381.
Foundation of Jiangsu Key Laboratory for the Research and Uti1ization of Plant Resources: SQ201401.

### Competing Interests

The authors declare there are no competing interests.

### Author Contributions

- Xiwu Qi performed the experiments, analyzed the data, wrote the paper, prepared figures and/or tables.
- Xu Yu and Daohua Xu performed the experiments.
- Hailing Fang analyzed the data.

- Ke Dong contributed reagents/materials/analysis tools.
- Weilin Li and Chengyuan Liang conceived and designed the experiments, reviewed drafts of the paper.

## DNA Deposition

The following information was supplied regarding the deposition of DNA sequences:

The *LjC4H3* and *LjC3H2* sequences described here are accessible via GenBank accession numbers KX845341 and KX845342.

## Data Availability

The raw data used for transcriptomic assemblies was downloaded from NCBI SRA with accession numbers SRR290309, SRR342027, SRR576924, SRR576925 and SRR766791 (https://ftp.ncbi.nih.gov/sra/sra-instant/reads/ByRun/sra/SRR/SRR290/SRR290309/SRR290309.sra

https://ftp.ncbi.nih.gov/sra/sra-instant/reads/ByRun/sra/SRR/SRR342/SRR342027/SRR342027.sra

https://ftp.ncbi.nih.gov/sra/sra-instant/reads/ByRun/sra/SRR/SRR576/SRR576924/SRR576924.sra

https://ftp.ncbi.nih.gov/sra/sra-instant/reads/ByRun/sra/SRR/SRR576/SRR576925/SRR576925.sra

https://ftp.ncbi.nih.gov/sra/sra-instant/reads/ByRun/sra/SRR/SRR766/SRR766791/SRR766791.sra).

## Supplemental Information

Supplemental information for this article can be found online at http://dx.doi.org/10.7717/peerj.3781#supplemental-information.

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
