# Peer review of "Identification and analysis of CYP450 genes from transcriptome of Lonicera japonica and expression analysis of chlorogenic acid biosynthesis related CYP450s"

_PeerJ, doi:10.7717/peerj.3781_

## Round 0.1 · original submission · Minor Revisions

I have carefully read through the manuscript and the two reviews and am recommending minor revisions for the manuscript. Overall the work in the manuscript is scientifically sound. Please address each comment from the two reviewers and make the necessary revisions as recommended.

Reviewer 1 ·

Basic reporting

no comment

Experimental design

no comment

Validity of the findings

no comment

Additional comments

Lonicera japonica is an important medicinal plant, with more than 500 traditional Chinese medicine prescriptions containing L. japonica. The active compounds of L. japonica include essential oils, phenolic acids, flavone, triterpenoid saponins, iridoilds and inorganic elements. Biosynthesis of almost all the secondary metabolites has cytochrome P450-dependent catalytic reaction. In this study, the authors identified 151 putative CYP450s from L. japonica. These 151 putative CYP450s were systematic analysis by classification, phylogenetic analysis and gene ontology classification. Among them, 47 CYP450s were totally assigned to 25 KEGG pathways, which might be involved in secondary metabolism. As CGA is the major component of L. japonica, CYP450s that might be involved in CGA biosynthesis were identified. Three C4H and two C3H were identified and cloned for functional related biosynthesis of CGA. The results showed that only LjC3H2 exhibited a similar expression pattern with accumulation of CGA. The authors implied that as C3H and C4H not only participated in CGA biosynthesis, but were also involved in other metabolites, such as flavonoids, anthocyanins, condensed tanins. This resulted in difference between the expression pattern of genes and accumulation of CGA.

In all, the manuscript was well organized. There are still several questions need to be clarified,
1. line 236, the sentence “Results indicated that all 151 CYP450s were mapped to one or more GO terms, of which 145 were assigned to the “cellular component”, 151 to the “molecular function”, and 151 to the “biological process” (Fig. 3).” is sophisticated. How to explain it?
2. Five CYP450s candidate for biosynthesis of CGA were analyzed by qRT-PCR, however, no close relationship was detected between the expression level and the accumulation of CGA. I think it might be useful to analyze the whole pathway genes as well with co-expression analysis to further understanding the expression pattern of these genes.
3. Knock-down or knock-out analysis is needed for confirming the function of these candidate CYP450s in biosynthesis of CGA in planta.

Reviewer 2 ·

Basic reporting

Clear, unambiguous, professional English language used throughout.
Intro & background to show context.
Literature well referenced & relevant.
Structure conforms to PeerJ standards, discipline norm, or improved for clarity.
Figures are relevant, high quality, well labelled & described.
Raw data supplied.

Experimental design

Original primary research within Aims and Scope of the journal.
Research question well defined, relevant & meaningful. It is stated how research fills an identified knowledge gap.

Validity of the findings

Conclusions are well stated, linked to original research question & limited to supporting results.

Additional comments

In RNA extraction and qRT‑PCR section, I can not see wether the authors have 3 biological replicates for qRT‑PCR analysis or not. This should be explained and corrected before Acceptance.

---

## Round 0.2 · accepted · Accept

I have looked over the revisions and believe the authors have adequately addressed the reviewer comments and recommend that the manuscript should be published.